# Plan-on-Graph: Self-Correcting Adaptive Planning of Large Language Model on Knowledge Graphs

**Liyi Chen**[1,2†]**, Panrong Tong**[2]**, Zhongming Jin**[2]**, Ying Sun**[3]**, Jieping Ye**[2*]**, Hui Xiong**[3,4*]

[1] University of Science and Technology of China, [2] Alibaba Cloud Computing,
[3] Thrust of Artificial Intelligence, The Hong Kong University of Science and
Technology (Guangzhou), [4] Department of Computer Science and Engineering,
The Hong Kong University of Science and Technology

`liyichencly@gmail.com, yings@hkust-gz.edu.cn, xionghui@ust.hk,`
`{panrong.tpr, zhongming.jinzm, yejieping.ye}@alibaba-inc.com`

## Abstract

Large Language Models (LLMs) have shown remarkable reasoning capabilities on complex tasks, but they still suffer from out-of-date knowledge, hallucinations, and opaque decision-making. In contrast, Knowledge Graphs (KGs) can provide explicit and editable knowledge for LLMs to alleviate these issues. Existing paradigm of KG-augmented LLM manually predefines the breadth of exploration space and requires flawless navigation in KGs. However, this paradigm cannot adaptively explore reasoning paths in KGs based on the question semantics and self-correct erroneous reasoning paths, resulting in a bottleneck in efficiency and effect. To address these limitations, we propose a novel self-correcting adaptive planning paradigm for KG-augmented LLM named Plan-on-Graph (PoG), which first decomposes the question into several sub-objectives and then repeats the process of adaptively exploring reasoning paths, updating memory, and reflecting on the need to self-correct erroneous reasoning paths until arriving at the answer. Specifically, three important mechanisms of *Guidance*, *Memory*, and *Reflection* are designed to work together, to guarantee the adaptive breadth of self-correcting planning for graph reasoning. Finally, extensive experiments on three real-world datasets demonstrate the effectiveness and efficiency of PoG.

## 1 Introduction

Large Language Models (LLMs) have manifested outstanding performance in various natural language processing and data science tasks, such as question answering [42, 25, 62], text generation [18, 13, 8, 15], recommender systems [60, 59, 51, 44], and domain-specific applications [46, 45, 14, 50, 36]. They leverage advanced deep learning techniques and immense amounts of pre-existing text data to understand and generate human language with impressive fluency and coherence. Despite their success in numerous applications, LLMs still suffer from out-of-date knowledge, hallucinations, and opaque decision-making, highlighting the ongoing need for further investigation in this rapidly evolving field.

Intuitively, as large-scale structural knowledge bases, Knowledge Graphs (KGs) [5, 1, 12, 43] provide explicit and editable depictions of massive real-world knowledge, which have the potential to be a promising complement to the drawbacks of LLMs. Previous studies manage to integrate KGs into LLM pre-training [61, 47] or fine-tuning [53, 27] stage. However, these methods mainly compress structured knowledge in KGs into LLMs' parameters in a black-box fashion and still cannot fully

---

†This work was accomplished when Liyi Chen was an intern at Alibaba Cloud Computing.
*Corresponding authors.

38th Conference on Neural Information Processing Systems (NeurIPS 2024).

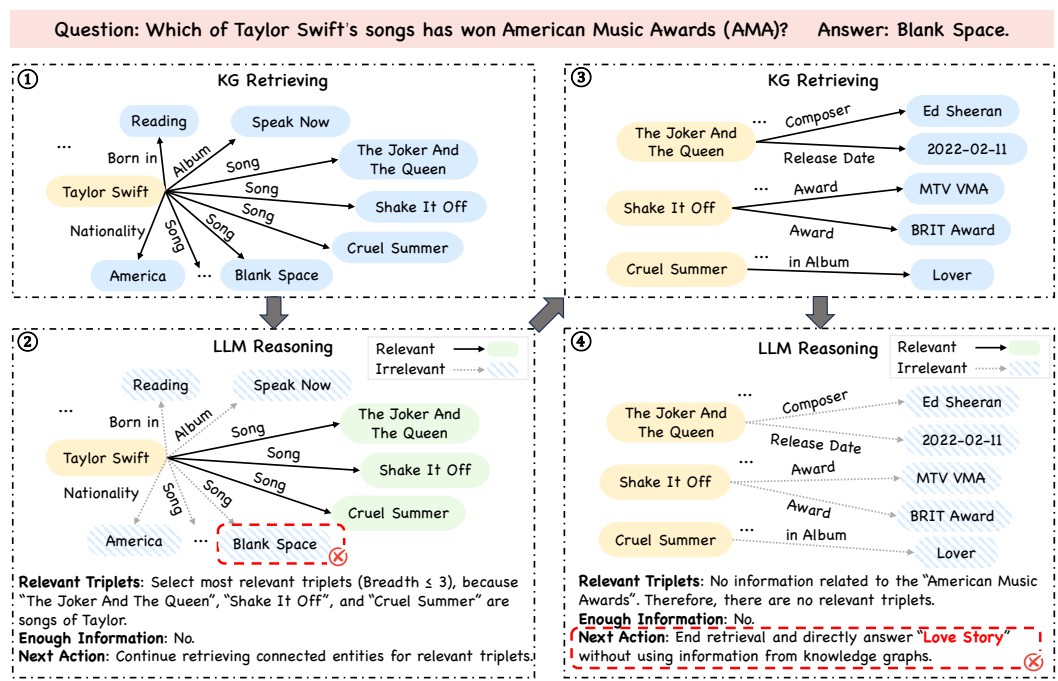

Figure 1: A toy example of existing KG-augmented LLM paradigm.

enhance the flexibility, reliability, and transparency of LLMs. Therefore, several attempts [3, 2, 40] first retrieve information from KGs and then deliver explicit knowledge into LLMs. Under these circumstances, LLMs do not directly participate in the graph reasoning process, making these methods excessively dependent on the KG completeness. Recently, a KG-augmented LLM paradigm has been proposed to conduct graph reasoning, which treats the LLM as an agent to interactively explore related entities and relations on KGs and perform reasoning based on the retrieved knowledge. For instance, StructGPT [19] and ToG [35] predefine the breadth of reasoning paths explored on the KG, and leverage the LLM to iterate the process of unidirectionally extending along the reasoning paths relevant to the question and reasoning the answer using these reasoning paths. This KG-augmented LLM paradigm offers an opportunity for more comprehensively amalgamating the knowledge from both the KG and LLM by facilitating the step-by-step derivation of further insights.

However, existing paradigm may fail to plan the exploration of correct reasoning paths for many complex questions. Figure 1 illustrates an example of existing paradigm's limitations when answering the question "Which of Taylor Swift's songs has won American Music Awards? (AMA)". These limitations lie in: (1) *Predefined path breadth*: Existing paradigm requires manually setting the breadth of reasoning paths in KGs, and a fixed breadth may result in all the selected relations or entities being incorrect. When determining the relevance between paths and questions in step ②, due to the limited maximum breadth of three and the uncertainty surrounding the respective awards of songs, the LLM selected maximum numbers of songs, ignoring the correct entity "Blank Space". (2) *Irreversible exploration direction*: The path exploration in existing paradigm is unidirectional without the ability to self-correct. Even if the paths are incorrect, the LLM still continues to extend current incorrect paths and lead to the failure of reasoning on the KG. In step ③ and ④, since "The Joker And The Queen", "Shake It Off", and "Cruel Summer" were already chosen, the reasoning process continued on incorrect paths and the right answer was not found. (3) *Forgetting partial conditions*: During reasoning, the LLM may forget partial conditions in the question and cannot provide the answer that satisfies multiple conditions simultaneously. In step ④, the LLM only remembered the condition that the song was by Taylor Swift but forgot the condition about the song winning an AMA award, leading to an incorrect answer, "Love Story". Therefore, the reasoning of complex questions may heavily rely on adaptive exploration and self-correction of erroneous reasoning paths.

To address these limitations, we propose a novel self-correcting adaptive planning paradigm for KG-augmented LLM named **P**lan-**o**n-**G**raph (**PoG**). To the best of our knowledge, we are the first to design a reflection mechanism for self-correction and adaptive KG exploration into KG-augmented LLMs, effectively improving the ability and efficiency of LLM reasoning. Specifically, PoG first decomposes

the question into several sub-objectives as guidance for planning exploration, and then repeats the process of adaptively exploring reasoning paths to access relevant KG data, updating memory to provide dynamic evidence for reflection, and reflecting on the need to self-correct reasoning paths until arriving at the answer. In PoG, three mechanisms are designed for adaptive self-correcting planning: (1) **Guidance**: To better guide adaptive exploration by harnessing conditions in the question, we employ the LLM to decompose the question into sub-objectives containing conditions, thereby benefiting the identification of relevant paths to each condition with flexible exploration breadth. (2) **Memory**: The information stored in memory offers historical retrieval and reasoning information for reflection. We record and update the *subgraph* to provide the LLM with all retrieved entities for initializing new exploration and self-correcting paths, *reasoning paths* to preserve the relationships between entities for LLM reasoning and allow for path correction, and *sub-objective status* to make the LLM recognize the known information of each condition and mitigate its forgetting in reflection stage. (3) **Reflection**: To determine whether to continue or self-correct current reasoning paths, we design a reflection mechanism to employ the LLM to reason whether to consider other entities into new exploration and decide which entities to backtrack to for self-correction based on information in memory. Finally, extensive experiments on three real-world KGQA datasets validate the effectiveness and efficiency of PoG [1]. The main contributions of this paper are listed as follows:

- We propose a novel self-correcting adaptive planning paradigm for KG-augmented LLM named PoG, which exploits the LLM to plan the adaptive breadth of reasoning paths and reflect to self-correct erroneous paths. To the best of our knowledge, we are the first to incorporate a reflection mechanism for self-correction and adaptive KG exploration into KG-augmented LLMs, effectively augmenting the LLM's reasoning ability.

- We specially design Guidance, Memory, and Reflection mechanisms for PoG. Guidance harnesses question conditions to better plan adaptive exploration by decomposing task into sub-objectives including conditions. Memory records the subgraph, reasoning paths, and sub-objective status to provide historical retrieval and reasoning information for Reflection. Based on Memory, Reflection reasons whether to self-correct reasoning paths and which entity to backtrack to for initiating new exploration.

- We conduct extensive experiments on three real-world KGQA datasets, namely CWQ, WebQSP, and GrailQA. The results demonstrate not only the effectiveness but also the efficiency of our proposed novel PoG paradigm for KG-augmented LLM.

## 2 Preliminary

**Knowledge Graph (KG)** stores massive factual knowledge in the form of a set of triplets: $G = \{(e, r, e') \,|\, e, e' \in E, r \in R\}$, where $E$ and $R$ denote the set of entities and relations, respectively.

**Relation Paths** are a sequence of relations: $z = \{r_1, r_2, ..., r_l\}$, where $r_i \in R$ denotes the $i$-th relation in the path and $l$ denotes the length of the path.

**Reasoning Paths** are the instances of a relation path $z$ in the KG: $p_z = e_0 \rightarrow r_1 e_1 \rightarrow r_2 e_2 \rightarrow ... \rightarrow r_l e_l$, where $e_i \in E$ denotes the $i$-th entity and $r_i$ denotes the $i$-th relation in the relation path $z$.

**Knowledge Graph Question Answering (KGQA)** is the task of answering natural language questions based on a set of facts over the KG. Given a question $q$, a knowledge graph $G$, and topic entities $T_q$ mentioned in $q$, the target of KGQA is to generate answers $A_q$ to the question $q$. Following previous studies [35], we assume any entity $e_q \in T_q$ mentioned in $q$ and answers $a_q \in A_q$ are labeled and linked to the corresponding entities in $G$, i.e., $T_q, A_q \subseteq E$.

## 3 Methodology

In this section, we introduce the technical details of the novel self-correcting adaptive planning paradigm for KG-augmented LLM named Plan-on-Graph (PoG). As illustrated in Figure 2, PoG consists of four key components: Task Decomposition, Path Exploration, Memory Updating, and Evaluation. PoG first decomposes the question into several sub-objectives as guidance of planning exploration and then repeats the process of adaptively exploring reasoning paths to access relevant KG data, updating memory to provide historical retrieval and reasoning information for reflection, and reflecting on the need to self-correct reasoning paths until arriving at the answer.

---

[1] https://github.com/liyichen-cly/PoG

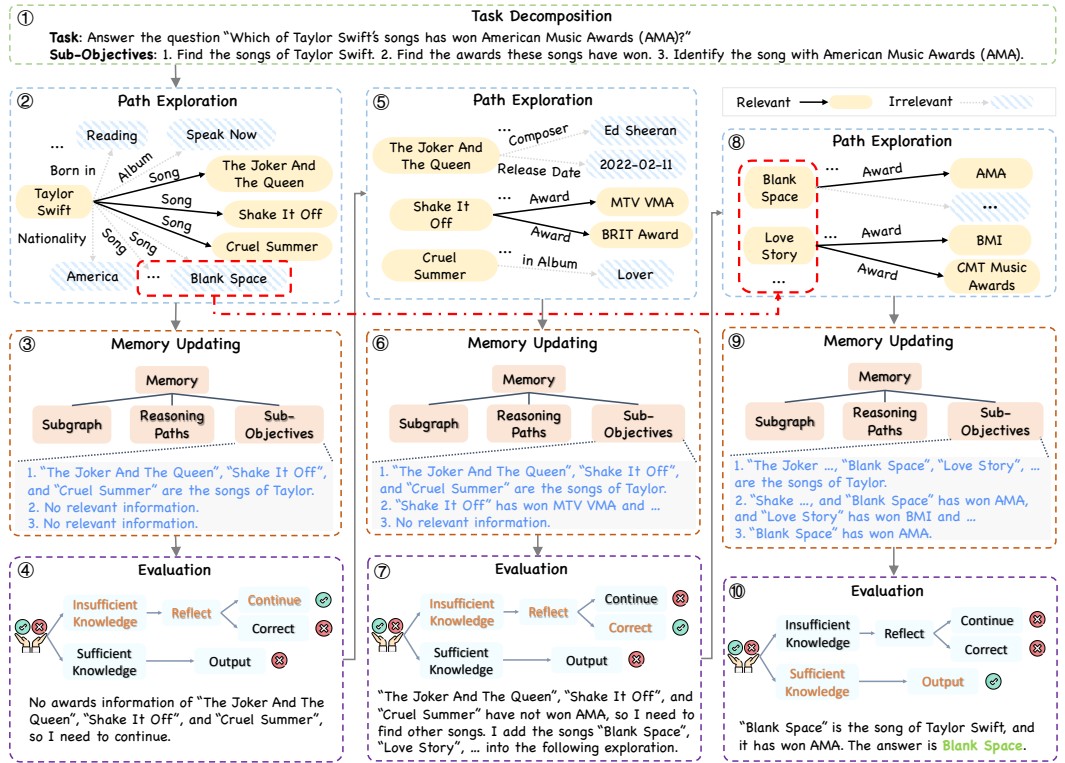

Figure 2: The framework overview of PoG, which includes four key components: Task Decomposition, Path Exploration, Memory Updating, and Evaluation.

## 3.1 Task Decomposition

To harness conditions in the question to better guide the adaptive exploration process, PoG decomposes the task of answering the question into multiple sub-objectives containing conditions through semantic analysis of the LLM. Sub-objectives serve as guidance for path exploration, benefiting the identification of relevant paths to each condition outlined in the question with flexible exploration breadth. Specifically, we prompt the LLM to decompose the original question $q$ into a list of sub-objectives for KG retrieval and reasoning. The prompt is shown in Appendix A.1. The list of sub-objectives can be denoted as $O = \{o_1, o_2, o_3, ...\}$. It is important to note that sub-objectives in $O$ may refer to the results obtained from other sub-objectives in $O$, allowing for interdependencies in the reasoning process.

## 3.2 Path Exploration

We access relevant information from the KG by exploring reasoning paths in the KG. On the initiation of path exploration, we localize the initial entities of reasoning paths, which correspond to the topic entities mentioned in the given question. Similar to prior research [19, 35], topic entities have been pre-identified and are part of the annotated datasets. Specifically, when presented with a question $q$, we use topic entities to serve as the initial elements of the reasoning paths, $E^0 = T_q = \{e_1^0, e_2^0, ..., e_{N_0}^0\}$, where $N_0$ is the number of topic entities.

In the subsequent iterations, we continue exploring reasoning paths most relevant to the question and suspend other reasoning paths. Taking the $D$-th iteration as an example, before the iteration starts, each reasoning path $p_n \in P$ consists of $D_{p_n}(D_{p_n} \leq D - 1)$ triplets, i.e., $p_n = \{(e_{s,n}^d, r_{j,n}^d, e_{o,n}^d)\}_{d=1}^{D_{p_n}}$, where $e_{s,n}^d$ and $e_{o,n}^d$ denote subject and object entities, $r_{j,n}^d$ is a specific relation between them, $(e_{s,n}^d, r_{j,n}^d, e_{o,n}^d)$ and $(e_{s,n}^{d+1}, r_{j,n}^{d+1}, e_{o,n}^{d+1})$ are linked to each other. It is noted that the length of each reasoning path may vary, because in the $D$-th iteration, we only continue exploring the reasoning paths most semantically relevant to the question, which are identified in the $D - 1$-th iteration. The sets of tail entities and relations to be explored are denoted as $E^{D-1} = \{e_1^{D-1}, e_2^{D-1}, ..., e_{N_{D-1}}^{D-1}\}$ and $R^{D-1} = \{r_1^{D-1}, r_2^{D-1}, ..., r_{N_{D-1}}^{D-1}\}$, respectively, where $N_{D-1}$ is the length of $E^{D-1}$ and

$R^{D-1}$. We leverage the LLM to identify the most relevant entities $E^D$ from the neighboring entities of the current entity set $E^{D-1}$ based on the question $q$ and extend the reasoning paths $P$ with $E^D$. In order to manage the complexity of dealing with a large number of neighboring entities using the LLM, we propose an adaptive exploration strategy that is not limited by the fixed number of relations and entities. This strategy involves a two-step process of finding relevant relations and utilizing these selected relations to explore entities.

**Relation Exploration.** Relation exploration is a process to retrieve the relations of all tail entities in $E^{D-1}$ and identify the most relevant relations to the question $q$ and the sub-objectives $O$. To be specific, we first conduct the search to obtain all relations linked to the tail entities in $E^{D-1}$ as the candidate relation set $R_{cand}^D = \{r_{cand,1}^D, r_{cand,2}^D, ..., r_{cand,N_{D-1}}^D\}$. We utilize $R_{cand}^D$ to extend the reasoning paths into candidate reasoning paths $P_{cand}$. Then, we employ the LLM to select a flexible number of relevant reasoning paths $P$ ending with the tail relations in $R^D$ from $P_{cand}$, based on the semantic information of the question $q$, tail entities $E^{D-1}$, candidate relations $R_{cand}^D$, and sub-objectives $O$. The prompt is shown in Appendix A.2.1, and the pre-defined query for relation search is shown in Appendix B.1.

**Entity Exploration.** Analogously, entity exploration is a process to retrieve neighboring entities based on $R^D$ and $E^{D-1}$ and detect the most relevant entities to the question $q$. From the previous relation exploration, we obtain extended reasoning paths $P$ and new tail relations $R^D$. For each reasoning path $p_n \in P$, we can execute the queries of $(e_n^{D-1}, r_n^D, ?)$ or $(?, r_n^D, e_n^{D-1})$ to retrieve a candidate entity set $E_{cand,n}^D$, where $e_n^{D-1}$ and $r_n^D$ are the tail entity and relation in $p_n$. When confronted with a large number of candidate entities, we use a small pre-trained DistilBERT [31] [2], to calculate the similarity between candidate entities and the question for recall. Then, we summarize all candidate entity sets into $E_{cand}^D$ and use $E_{cand}^D$ as the tail entities to expand $P$ into $P_{cand}$. With the candidate reasoning paths $P_{cand}$, we exploit the LLM to choose a flexible number of relevant reasoning paths $P$ ending with the tail entities $E^D$ from $P_{cand}$, based on the semantic information of the question $q$ and knowledge triplets composed of tail entities $E^{D-1}$, tail relations $R^D$ and candidate entities $E_{cand}^D$. The prompt is shown in Appendix A.2.2, and the pre-defined query for entity search is shown in Appendix B.2.

## 3.3 Memory Updating

The information stored in memory provides historical retrieval and reasoning information for reflection. After a two-step exploration, we dynamically update the searched subgraph $G_{Sub}$, reasoning paths $P$, and sub-objective status $S$ in memory based on the ongoing reasoning process.

**Subgraph.** The subgraph includes all retrieved relations and entities from the KG. We update the subgraph in memory, which can be utilized during later reflection to determine which entity to backtrack to for self-correction. In the $D$-th iteration, the searched subgraph $G_{Sub}$ is updated by adding the retrieved candidate relation set $R_{cand}^D$ and candidate entity set $E_{cand}^D$.

**Reasoning Paths.** In order to ensure that the LLM can understand relationships between entities for better reasoning and allow for path correction in reflection stage, we update reasoning paths $P$ to preserve the semantic structure within the KG.

**Sub-Objective Status.** The LLM may forget partial conditions in the reasoning process. Sub-objectives obtained by decomposing the question can help the LLM remember multiple conditions in the question. The status of sub-objectives contains the current known information related to the sub-objectives, which can aid the LLM in remembering the known information of each condition and determining whether to correct the exploration direction in reflection stage. Hence, we leverage the LLM to update the currently known information relevant to sub-objectives into sub-objective status $S = \{s_1, s_2, s_3, ...\}, |S| = |O|$, based on the semantic information of the question $q$, sub-objectives $O$, historical sub-objective status, and reasoning paths $P$, along with the LLM's own knowledge. The prompt is shown in Appendix A.3.

## 3.4 Evaluation

After the path exploration and memory updating, PoG prompts the LLM to reason whether the current acquired information, including sub-objective states and reasoning paths recorded in memory, is sufficient to infer an answer. The prompt is shown in Appendix A.4.1. If the LLM determines that the

---

[2] https://huggingface.co/sentence-transformers/msmarco-distilbert-base-tas-b

information is sufficient, it will integrate reasoning paths, sub-objective states, and its own knowledge to provide an answer. When information is considered insufficient, there may be two situations. One is that PoG will acquire sufficient information after further extension of current paths, and the other is that current paths are incorrect. Since the reasoning capability of the LLM does not always guarantee the correctness of path exploration, there is a need to self-correct erroneous reasoning paths. Therefore, we design a reflection mechanism to determine whether and how to self-correct reasoning paths. When the LLM believes that the information is insufficient, PoG enters the stage of reflection. Specifically, PoG utilizes the LLM to reflect on whether to correct the current exploration direction based on the question $q$, sub-objective status $S$, reasoning paths $P$, and entities planned for the next iteration of retrieval $E^D$ from memory. Besides, the LLM will provide the reason for the reflection result. If the LLM judges that it is necessary to incorporate additional entities beyond those in $E^D$ for exploration, then a self-correction of reasoning paths is needed. Otherwise, PoG will continue exploring along the current reasoning paths with tail entities in $E^D$. For self-correction, PoG employs the LLM to decide which entities in $E_{\text{cand}} = E_{\text{cand}}^1 \cup E_{\text{cand}}^2 \cup ... \cup E_{\text{cand}}^D$ to backtrack to based on sub-objective states in $S$ and the reason for additional retrieval obtained from the reflection, and adds new exploration of backtracked entities $E_{\text{add}}^D$ into $E^D$ for the self-correction, denoted as $E^D = E^D \cup E_{\text{add}}^D$. The prompts for reflection are shown in Appendix A.4.2.

## 4 Experiments

### 4.1 Experimental Setups

#### 4.1.1 Datasets & Evaluation Metrics

To demonstrate the effectiveness of PoG on complex reasoning over knowledge graphs, we adopt three representative multi-hop KGQA datasets: CWQ [37], WebQSP [56], and GrailQA [17]. All three datasets rely on the external knowledge graph from Freebase [5]. For the large dataset GrailQA, we utilize the same testing samples as those in ToG [35] to improve computational efficiency. Following prior research [23, 19, 35], we use exact match accuracy (Hits@1) as the evaluation metric.

#### 4.1.2 Comparison Methods

Due to variations in the performance of the method across different datasets, we select prior state-of-the-art (SOTA) approaches as baselines for each dataset. They can be categorized into two groups: (1) *LLM-only methods*, including standard prompting (IO prompt) [6], Chain-of-Thought prompting (CoT) [49], and Self-Consistency (SC) [48]. (2) *KG-augmented LLM methods*, including fine-tuned and prompting methods. For CWQ and WebQSP, we utilize UniKGQA [20], TIARA [34], RE-KBQA [7], DeCAF [57], and RoG [27] as fine-tuned baselines and KD-CoT [41], KB-BINDER [23], StructGPT [19], Interactive KBQA [52], and ToG [35] as prompting baselines. For GrailQA, we utilize RnG-KBQA [55], TIARA [34], FC-KBQA [58], Pangu [16], FlexKBQA [26], and GAIN [33] as fine-tuned baselines and KB-BINDER [23] and ToG [35] as prompting baselines. The descriptions of baselines are presented in Appendix D.

Table 1: Performance comparison of different methods on CWQ and WebQSP.

| Method | CWQ | WebQSP |
|---|---|---|
| *LLM-Only* | | |
| IO Prompt [6] | 37.6 | 63.3 |
| CoT [49] | 38.8 | 62.2 |
| SC [48] | 45.4 | 61.1 |
| *Fine-Tuned KG-Augmented LLM* | | |
| UniKGQA [20] | 51.2 | 79.1 |
| TIARA [34] | - | 75.2 |
| RE-KBQA [7] | 50.3 | 74.6 |
| DeCAF [57] | 70.4 | 82.1 |
| RoG [27] | 62.6 | 85.7 |
| *Prompting KG-Augmented LLM w/GPT-3.5 or others* | | |
| KD-CoT [41] | 50.5 | 73.7 |
| KB-BINDER [23] | - | 74.4 |
| StructGPT [19] | 54.3 | 72.6 |
| ToG [35] | 57.1 | 76.2 |
| **PoG** | **63.2** | **82.0** |
| *Prompting KG-Augmented LLM w/GPT-4* | | |
| InteractiveKBQA [52] | 59.2 | 72.5 |
| ToG [35] | 67.6 | 82.6 |
| **PoG** | **75.0** | **87.3** |

### 4.2 Performance Comparison

We compare PoG with the SOTA baselines to demonstrate its effectiveness for KG-augmented LLM. Table 1 and Table 2 present the experimental results on CWQ, WebQSP, and GrailQA datasets. Overall, PoG achieves the best performance across all three datasets. Specifically, we can make the following observations. First, compared to all prompting KG-augmented LLM baselines, PoG shows superior performance advantages. Regardless of whether GPT-3.5 or

Table 2: Performance comparison of different methods on GrailQA.

| Method | GrailQA | | | |
|---|---|---|---|---|
| | Overall | I.I.D. | Compositional | Zero-shot |
| *LLM-Only* | | | | |
| IO Prompt [6] | 29.4 | - | - | - |
| CoT [49] | 28.1 | - | - | - |
| SC [48] | 29.6 | - | - | - |
| *Fine-Tuned KG-Augmented LLM* | | | | |
| RnG-KBQA [55] | 68.8 | 86.2 | 63.8 | 63.0 |
| TIARA [34] | 73.0 | 87.8 | 69.2 | 68.0 |
| FC-KBQA [58] | 73.2 | 88.5 | 70.0 | 67.6 |
| Pangu [16] | 75.4 | 84.4 | 74.6 | 71.6 |
| FlexKBQA [26] | 62.8 | 71.3 | 59.1 | 60.6 |
| GAIN [33] | 76.3 | 88.5 | 73.7 | 71.8 |
| *Prompting KG-Augmented LLM w/GPT-3.5 or others* | | | | |
| KB-BINDER [23] | 50.6 | - | - | - |
| ToG [35] | 68.7 | 70.1 | 56.1 | 72.7 |
| **PoG** | **76.5** | **76.3** | **62.1** | **81.7** |
| *Prompting KG-Augmented LLM w/GPT-4* | | | | |
| ToG [35] | 81.4 | 79.4 | 67.3 | 86.5 |
| **PoG** | **84.7** | **87.9** | **69.7** | **88.6** |

GPT-4 is used as the underlying LLM, PoG substantially outperforms the SOTA baseline, ToG. ToG explores reasoning paths with a fixed exploration breadth and cannot detect or correct the errors, showing limitations in effect and efficiency. Meantime, we specially design self-correction and adaptive planning mechanisms, which can effectively improve both performance and efficiency. Second, although PoG is a training-free prompting method, its performance is highly competitive with fine-tuned KG-augmented LLM baselines. When using GPT-4, the performance of PoG exceeds all fine-tuned KG-augmented LLM baselines across the board. Even with GPT-3.5, the result of PoG on GrailQA surpasses all fine-tuned KG-augmented LLM methods. This suggests that our designed guidance, memory, and reflection mechanisms allow PoG's effect to surpass most of the fine-tuned methods. Third, the improvement of PoG is obvious when compared to LLM-only baselines, which do not leverage external KGs. Besides, all KG-augmented LLM methods consistently outperform LLM-only methods, indicating the value of incorporating KGs to enhance LLM performance. Moreover, PoG further improves the effectiveness of KG-augmented LLMs through its self-correctable adaptive planning. Additionally, it is worth noting that PoG with GPT-3.5 outperforms other methods on the zero-shot subset of the GrailQA dataset by a large margin, apparently outperforming all fine-tuned KG-augmented LLMs on this category. The self-correction mechanism in PoG allows it to dynamically correct errors during the reasoning process, which is crucial for zero-shot problems.

### 4.3 Ablation Study

In order to assess the effectiveness of each mechanism and adaptive exploration in PoG, we conduct the ablation study to remove them on three datasets, respectively. Specifically, w/o Guidance refers to the variant where entire task decomposition as guidance is removed. w/o Memory indicates the variant without the memory mechanism. w/o Reflection refers to the variant

Table 3: Performance of removing each mechanism and adaptive exploration, respectively.

| Method | CWQ | WebQSP | GrailQA |
|---|---|---|---|
| **PoG** | **63.2** | **82.0** | **76.5** |
| w/o Guidance | 60.1 | 80.3 | 72.4 |
| w/o Memory | 58.9 | 77.5 | 69.3 |
| w/o Reflection | 59.4 | 78.1 | 70.5 |
| w/o Adaptive Breadth | 61.3 | 80.2 | 73.8 |

where, in the case of insufficient information, it only continues exploring along the original reasoning paths. w/o Adaptive Breadth means that the variant uses a fixed exploration space breadth instead of adapting it based on the situation. Table 3 shows the performance of all variants and the results suggest that each mechanism and adaptive breadth appears to contribute positively to the overall performance, with their removal leading to weaker results on complex question answering tasks across the evaluated datasets. These variations achieve a minimum reduction of 3.0%, 2.1%, and 3.5% on CWQ, WebQSP, and GrailQA, respectively. The performance of w/o Memory drops the

Table 4: Efficiency comparison between our proposed PoG and the baseline ToG.

| Dataset | Method | LLM Call | Input Token | Output Token | Total Token | Time (s) |
|---------|--------|----------|-------------|--------------|-------------|----------|
| CWQ | ToG | 22.6 | 8,182.9 | 1,486.4 | 9,669.4 | 96.5 |
| | **PoG** | **13.3** | **7,803.0** | **353.2** | **8,156.2** | **23.3** |
| WebQSP | ToG | 15.9 | 6,031.2 | 987.7 | 7,018.9 | 63.1 |
| | **PoG** | **9.0** | **5,234.8** | **282.9** | **5,517.7** | **16.8** |
| GrailQA | ToG | 11.1 | 4,066.0 | 774.6 | 4,840.6 | 50.2 |
| | **PoG** | **6.5** | **3,372.8** | **202.8** | **3,575.6** | **11.5** |

most, followed by w/o Reflection, because without the memory there is no information to support PoG in navigating the exploration and achieving self-correction, and PoG cannot self-correct the wrong reasoning paths without the reflection mechanism. Moreover, after setting a fixed maximum breadth for exploration, the performance deteriorates. This indicates that a fixed breadth makes the method lack flexibility and less adaptable to different questions. However, because the mechanisms of memory and reflection ensure that PoG is able to self-correct, the performance of w/o Adaptive Breadth does not decrease a lot.

## 4.4 Efficiency Study

We study the efficiency of PoG and the SOTA prompting KG-augmented LLM baseline, ToG. Table 4 presents the average LLM call, token consumption, and time required by both methods to answer a question across three datasets. In all datasets, PoG demonstrates clear advantages over ToG in terms of all metrics. For average number of LLM calls, PoG consistently requires fewer calls to the LLM, and reduces it by at least 40.8%. This highlights PoG's ability to reason more efficiently with fewer LLM interactions. Regarding token consumption, PoG exhibits a notable advantage in both input and output token usage. On CWQ, compared to ToG's input tokens, PoG shows a reduction of approximately 4.6% in input token consumption. As for output tokens, PoG produces just 353.159 output tokens, representing a substantial decrease of roughly 76.2%. This indicates the effectiveness of PoG in reducing the overall token consumption during the reasoning process. Most importantly, PoG achieves superior time efficiency compared to ToG. On CWQ and GrailQA, PoG presents a speedup of over 4 times. ToG predefines the breadth of exploration, leading to the exploration of many irrelevant paths. Additionally, ToG lacks a self-correction mechanism, and when there is insufficient information to answer a question, it can only extend the current reasoning paths, sacrificing a lot of efficiency on irrelevant explorations. By contrast, the efficiency advantages of PoG can be attributed to its adaptive exploration and self-correction of reasoning paths based on the semantics of the question. The adaptive breadth reduces unnecessary exploration, and effective correction avoids the extending of wrong current paths.

## 4.5 Case Study

Figure 3 shows a typical case from the testing results on CWQ dataset. We compare the results of PoG, ToG, and CoT in answering the question "Who is in control of the place where the movie 'The Naked and the Dead' takes place?". The underlying LLMs they used are all based on GPT-3.5. PoG initially identifies a flexible number of relations related to the topic entities. Specifically, for "The Naked and the Dead", PoG successfully discovers that the movie takes place in Panama, while for "President of Panama", the LLM thinks that only the relation "government.government_office_or_title.jurisdiction" is relevant. Upon retrieval, no information is found regarding the person in control of Panama. This triggers reflection as PoG realizes that it lacks sufficient information. With the memory, PoG refers to the sub-objective status and recognizes that it already knows the movie location (Panama) for sub-objective #1 but is unaware of the person in control of Panama for sub-objective #2. Based on the current reasoning paths, PoG makes a decision to execute self-correction and returns to exploring the relation not previously explored for "President of Panama". Due to the task decomposition, during the self-correction process, it becomes easier to identify the correct relation "government.government_office_or_title.office_holders" according to the sub-objectives. Through the guidance, memory, and reflection mechanisms, PoG successfully finds the correct answer, "Juan Carlos Varela". In contrast, ToG fails to identify the most relevant relation concerning "President of Panama" and continues exploring incorrect paths. This consumes a significant amount of time and ultimately leads to an erroneous answer due to the hallucination. CoT refuses to answer directly since the LLM realizes its lack of knowledge regarding the answer and requires additional information to be provided. From this analysis, it is evident that PoG outperforms ToG and CoT. PoG successfully

| | **Question: Who is in control of the place where the movie "The Naked and the Dead" takes place?** |
|---|---|
| **CoT** | ***Answer***: [Refused to answer.] I'm sorry, but I need to clarify the question a bit further in order to provide an accurate answer. Could you please specify whether you are referring to the character or the entity in control of the place where the movie "The Naked and the Dead" takes place? Thank you for your understanding. |
| **ToG** | ***Answer***: [KG exploration failed, and the LLM answered the question according to its own knowledge.] First, the movie "The Naked and the Dead" takes place during World War II. Second, the United States Army is in control of the place where the movie takes place.
***Paths***: The Naked and the Dead → *film.film.country* → United States of America
↘ *film.film.featured_film_locations* → Panama ↖
President of Panama → *government.government_office_or_title.jurisdiction*
↘ ↘ *government.governmental_jurisdiction.government_positions* ⋯
*government.government_position_held.office_position_or_title* ⋯ |
| **PoG** | ***Answer***: [Through reflection to self-correct paths, the answer was found on the revised paths.] The movie "The Naked and the Dead" takes place in Panama. The President of Panama, Juan Carlos Varela, is in control of Panama based on the knowledge triplets.
***Sub-Objectives***: 1. Identify the place where the movie "The Naked and the Dead" takes place; 2. Determine the authority or governing body in control of that place.
***Sub-Objective Status***: 1. The movie "The Naked and the Dead" takes place in Panama; 2. The President of Panama, Juan Carlos Varela is in control of Panama.
***Paths***: The Naked and the Dead → *film.film.country* → United States of America
↘ *film.film.featured_film_locations* → Panama ↖
President of Panama → *government.government_office_or_title.jurisdiction*
↘ *government.government_office_or_title.office_holders* → m.010gg02t
Juan Carlos Varela ← *government.government_position_held.office_holder* ↙ |

Figure 3: A typical case to compare different methods to answer the complex question. For the convenience of display, we only provide the sub-objective status and partial reasoning paths stored in memory. Topic entities, wrong answers, and correct answers are highlighted in blue, red, and green. The revised path is highlighted with a yellow background.

leverages sub-objective status to self-correct the exploration path in the reflection stage and finally provides the correct answer.

## 5 Related Work

**LLM Reasoning.** To encourage LLMs to engage in reasoning rather than simply providing answers directly, many researchers instruct LLMs to generate the process of thinking in their outputs [49, 22, 63]. In the early stages, Chain of Thought (CoT) [49] was designed to provide a few examples of intermediate natural language reasoning steps as the prompt. After that, several variants of CoT reasoning with different forms like Tree-of-Thought [54], Graph-of-Thought [4], Memory of Thought [24], and Skeleton-of-Thought [30] were proposed to enhance the thinking process. However, LLMs may make mistakes during the reasoning process. Hence, many works [32, 21, 28, 29] designed self-correction mechanisms based on feedback to rectify flawed reasoning and ensure accuracy. Additionally, large efforts were dedicated to guiding LLMs in understanding complex graph structures [39, 38] and improving their graph reasoning across different graph tasks [11, 10, 9]. However, it is still an open issue to address the outdated knowledge, hallucinations, and opaque decision-making for LLM reasoning.

**KG-Augmented LLM.** Despite the pre-training of LLMs on massive corpora, they still suffer from limitations such as outdated knowledge, hallucinations, and opaque decision-making. An effective approach to address these limitations is to leverage KGs for explicit and editable knowledge provision to LLMs. Previous studies integrated KGs into LLM pre-training [61, 47] or fine-tuning [53, 27] stage, but they merely inject structured knowledge into LLMs' parameters and still leave these limitations unexplored. Therefore, several works [3, 2, 40] first retrieved information from KGs and then directly fed explicit knowledge into LLMs. In this way, LLMs do not involve the graph reasoning process and cannot provide potential insights. Then, a novel KG-augmented LLM paradigm [19, 35]

was proposed to treat the LLM as an agent to interactively explore related entities and relations on KGs and perform reasoning based on the retrieved knowledge. Although this KG-augmented LLM paradigm has achieved impressive performance, it still faces the challenges of adaptively exploring the KG based on question semantics and self-correcting erroneous reasoning paths. To the best of our knowledge, our work stands out as a pioneering effort in successfully integrating a reflection mechanism for self-correction and adaptive KG exploration into KG-augmented LLMs, effectively enhancing the LLM's reasoning ability.

## 6    Conclusion

In this paper, we proposed a novel self-correcting adaptive planning paradigm for KG-augmented LLM named Plan-on-Graph (PoG). To the best of our knowledge, we were the first to incorporate a reflection mechanism for self-correction and adaptive KG exploration into KG-augmented LLMs, effectively augmenting LLM's reasoning ability and efficiency. PoG first decomposed the question into several sub-objectives, and then repeated the process of exploring reasoning paths, updating memory, and reflecting on the need to self-correct reasoning paths until arriving at the answer. To be specific, three important mechanisms were designed to work together to guarantee the adaptive breadth of self-correcting planning for graph reasoning, i.e., Guidance, Memory, and Reflection. Finally, extensive experiments on three real-world KGQA datasets validated not only the effectiveness but also the efficiency of the proposed PoG.

### Acknowledgments and Disclosure of Funding

This work was supported in part by the National Key Research and Development Program of China (Grant No. 2023YFF0725001), the National Natural Science Foundation of China (Grant No. 92370204, 62306255), the Guangdong Basic and Applied Basic Research Foundation (Grant No. 2023B1515120057, 2024A1515011839), the Guangzhou-HKUST (GZ) Joint Funding Program (Grant No. 2023A03J0008), and the Alibaba Research Intern Program.

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

# Appendix

## A Prompts

Here, we provide all the prompts used in PoG. To facilitate the LLM output parsing, we require the LLM to provide answers using specific data structures, such as lists and JSON. Besides, we require the LLM not to output any other irrelevant information to the results. The specific in-context few-shot is shown in code files.

### A.1 Task Decomposition

```
Please break down the process of answering the question into as few sub-objectives
as possible based on semantic analysis.

In-Context Few-Shot

Now you need to directly output sub-objectives of the following question in list
format without other information or notes.
Q: {}
```

### A.2 Path Exploration

#### A.2.1 Relation Exploration

```
Please provide as few highly relevant relations as possible to the question and its
sub-objectives from the following relations (separated by semicolons).

In-Context Few-Shot

Now you need to directly output relations highly related to the following question
and its sub-objectives in list format without other information or notes.
Q: {}
Sub-Objectives: {}
Topic Entity: {}
Relations: {}
```

#### A.2.2 Entity Exploration

```
Which entities in the following list ([] in Triples) can be used to answer the
question? Please provide the minimum possible number of entities, and strictly
adhering to the constraints mentioned in the question.

In-Context Few-Shot

Now you need to directly output the entities from [] in Triplets for the following
question in list format without other information or notes.
Q: {}
Triplets: {}
```

### A.3 Memory Updating

```
Based on the provided information (which may have missing parts and require further
retrieval) and your own knowledge, output the currently known information required
to achieve the sub-objectives.

In-Context Few-Shot

Now you need to directly output the results of the following question in JSON format
 without other information or notes.
Q: {}
```

```
Sub-Objectives: {}
Memory: {}
Knowledge Triplets: {}
```

## A.4   Evaluation

### A.4.1   Answer Question

```
Please answer the question based on the memory, related knowledge triplets and your
knowledge.

In-Context Few-Shot

Now you need to directly output the results of the following question in JSON format
 (must include "A" and "R") without other information or notes. If the triplets
explicitly contain the answer to the question, prioritize the fact of the triplet
over memory.
Q: {}
Memory: {}
Knowledge Triplets: {}
```

### A.4.2   Reflection

```
Based on the current set of entities to be retrieved and the known information
including memory and triplets, is it necessary to add additional entities for
answering the question?

In-Context Few-Shot

Now you need to directly output the results of the following question in the JSON
format (must include "Add" and "Reason") without other information or notes.
Q: {}
Entities set to be retrieved: {}
Memory: {}
Knowledge Triplets: {}
```

```
Please select the fewest necessary entities to be retrieved for answering the Q from
 Candidate Entities, based on the current known information (Memory), the reason for
 additional retrieval, and your own knowledge.

In-Context Few-Shot

Now you need to directly output the results for the following Q in the list format
without other information or notes.
Q: {}
Reason: {}
Candidate Entities: {}
Memory: {}
```

## B   Search SPARQL

To automatically process the KG data in PoG, we pre-define the SPARQL for Freebase queries, which
can be executed by filling in the entity's mid and relation.

### B.1   Relation Search

```
PREFIX ns: <http://rdf.freebase.com/ns/>
SELECT DISTINCT ?relation
WHERE {
  ns:mid ?relation ?x .
```

```
}
```

```
PREFIX ns: <http://rdf.freebase.com/ns/>
SELECT DISTINCT ?relation
WHERE {
  ?x ?relation ns:mid .
}
```

## B.2 Entity Search

```
PREFIX ns: <http://rdf.freebase.com/ns/>
SELECT ?tailEntity
WHERE {
  ns:mid ns:relation ?tailEntity .
}
```

```
PREFIX ns: <http://rdf.freebase.com/ns/>
SELECT ?tailEntity
WHERE {
  ?tailEntity ns:relation ns:mid .
}
```

## B.3 Entity Name Search

```
PREFIX ns: <http://rdf.freebase.com/ns/>
SELECT DISTINCT ?tailEntity
WHERE {
  {
    ?entity ns:type.object.name ?tailEntity .
    FILTER(?entity = ns:mid)
  }
  UNION
  {
    ?entity <http://www.w3.org/2002/07/owl#sameAs> ?tailEntity .
    FILTER(?entity = ns:mid)
  }
}
```

## C  Datasets

In this paper, we use three complex multi-hop KGQA datasets: ComplexWebQuestions [37], WebQSP [56], and GrailQA [17]. The statistics of datasets are shown in Table 5. WebQSP contains questions from WebQuestions that are answerable by Freebase. It tests I.I.D. generalization on questions. ComplexWebQuestions (CWQ) extends WebQSP and encompasses four types of complex questions: conjunction, composition, comparative, and superlative. GrailQA is a diverse KGQA dataset built on Freebase, and is designed to test three levels of generalization of models: I.I.D., compositional, and zero-shot.

Table 5: Statistics of KGQA datasets.

| Dataset | Answer Format | Train | Test | Licence |
|---------|---------------|-------|------|---------|
| ComplexWebQuestions | Entity | 27,734 | 3,531 | - |
| WebQSP | Entity/Number | 3,098 | 1,639 | CC Licence |
| GrailQA | Entity/Number | 44,337 | 1,000 | - |

# D  Baseline Descriptions

The baselines we compare can be categorized into two groups: (1) LLM-only methods; (2) KG-augmented LLM methods, including fine-tuned and prompting methods.

## LLM-Only Methods

- Standard prompting (IO prompt) [6] verifies the ability of LLMs to achieve better performance in task-agnostic, few-shot problems than traditional LMs.

- Chain-of-Thought prompting (CoT) [49] generates a series of intermediate reasoning steps in prompts to help LLMs perform better in several NLP tasks.

- Self-Consistency (SC) [48] samples multiple, diverse reasoning paths through few-shot CoT, and uses the generations to select the most consistent answer.

## Finetuned KG-Augmented LLM Methods

- UniKGQA [20] unifies the graph retrieval and reasoning process into a single model with LLMs.

- TIARA [34] first uses BERT to retrieve a set of schema items, which are further used as the input, together with the question, to T5 for plan generation. They also apply constrained decoding but only for grammaticality.

- RE-KBQA [7] capitalizes relations in KGs to enhance entity representations and introduce additional supervision to improve the selection of reasoning paths.

- DeCAF [57] combines semantic parsing and LLMs reasoning to jointly generate answers, which also reach salient performance on KGQA tasks.

- RoG [27] collaborates LLMs with KGs to achieve trustworthy reasoning to leverage structural information.

- RnG-KBQA [55] first uses BERT to rank a set of enumerated candidate programs (up to a limited complexity), and then uses T5 to edit the top programs into more complex programs.

- FC-KBQA [58] proposes a fine-to-coarse composition framework to avoid knowledge entanglement and guarantee both generalization ability and logical interpretability.

- Pangu [16] considers leveraging the discriminative ability of LLMs. It consists of a symbolic agent with a cooperative neural LLM.

- FlexKBQA [26] is a flexible KGQA framework with LLMs. It can utilize a limited set of annotated data to build KGQA for different KGs and query languages.

- GAIN [33] pays attention to the robustness of KGQA models. It proposes a data augmentation method to alleviate this problem and further evaluates the distribution shifts including from different aspects.

## Prompting KG-Augmented LLM Methods

- KB-BINDER [23] is developed to challenge the heterogeneity of items from different KGs. It enables few-shot in-context learning over KGQA tasks.

- KD-CoT [41] retrieves relevant knowledge from KGs to generate faithful reasoning plans for LLMs.

- StructGPT [19] defines the interface of KG data to implement knowledge access and filtering with finite quantity, and leverage the LLM to infer the answer or subsequent planning repeatedly.

- Interactive KBQA [52] interacts with KGs directly and then generates logical forms. The interactions are under three designed universal APIs for KGs.

- ToG [35] iteratively retrieves relevant triplets from KGs and employs the LLM to assess whether the reasoning paths in beam search are sufficient for answering the question and if further retrieval of the next hop is necessary.

# E    Implementation Details

In our experiments, we use GPT-3.5 and GPT-4 to serve as the underlying LLMs. We call them by the OpenAI official API [3]. We set the temperature parameter to 0.3, frequency penalty to 0, and presence penalty to 0. The maximum token length for generation is 1024. In all experiments, the depth of exploration is set to 4 to avoid endless exploration. The experiments are conducted on a server with two Intel(R) Xeon(R) CPU E5-2682 v4 @ 2.50GHz and 256 GB RAM memory.

# F    Depth Sensitivity

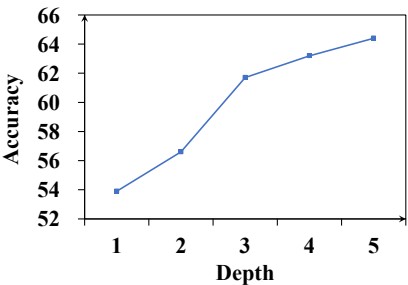

Since LLMs are not entirely certain about when to stop, we need to manually set the depth of KG exploration to avoid endless exploration. To investigate the impact of exploration depth on PoG performance, we conduct experiments with depth settings ranging from 1 to 5 on CWQ dataset. As shown in Figure 4, increasing the depth will improve the performance of PoG. Beyond a depth of 4, the improvement becomes less noticeable. The increase in depth leads to exponential growth in resource and time consumption. Considering the balance between efficiency and effectiveness, we set the depth to 4.

Figure 4: The impact of exploration depth on the performance of PoG.

# G    Case Analysis

In PoG, we design a reflection mechanism to provide the opportunity for self-correction for exploring reasoning paths. In Figure 5, we calculate the proportion of cases with reverse occurrences among all questions in CWQ, and the results show that 24% of cases involve reversing during the exploration process to achieve self-correction. This demonstrates that LLMs are indeed not always capable of making correct judgments in KG exploration and that self-correction is necessary for KG-augmented LLMs. Figure 6 presents the proportion of correct answers obtained by PoG after self-correction on three datasets. Overall, the self-correction in PoG appears to have positively impacted the accuracy of KGQA, particularly for the WebQSP and CWQ datasets, where the proportion of correct answers reached 64% and 48% after the self-correction process. This analysis suggests that the reflection mechanism in PoG has the potential to enhance the reasoning capabilities of KG-augmented LLM and improve the performance across various datasets by allowing for self-correction and exploration of alternative reasoning paths.

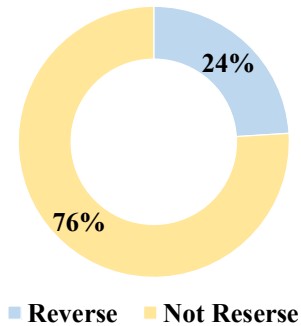

Figure 5:  The proportion of cases with reverse occurrences among all data.

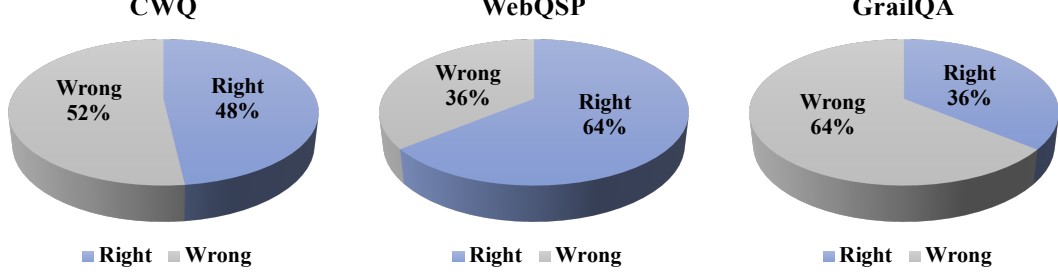

Figure 6: The proportion of correct answers obtained by PoG after self-correction.

Besides, Figure 7 shows another typical case from the testing results on CWQ dataset. We compare the results of PoG, ToG, and CoT in answering the question "What genre of music favored by Claude Debussy appears in the movie Suzanne Farrell: Elusive Muse?". PoG first adaptively identifies the

---
[3] https://platform.openai.com/docs/api-reference.

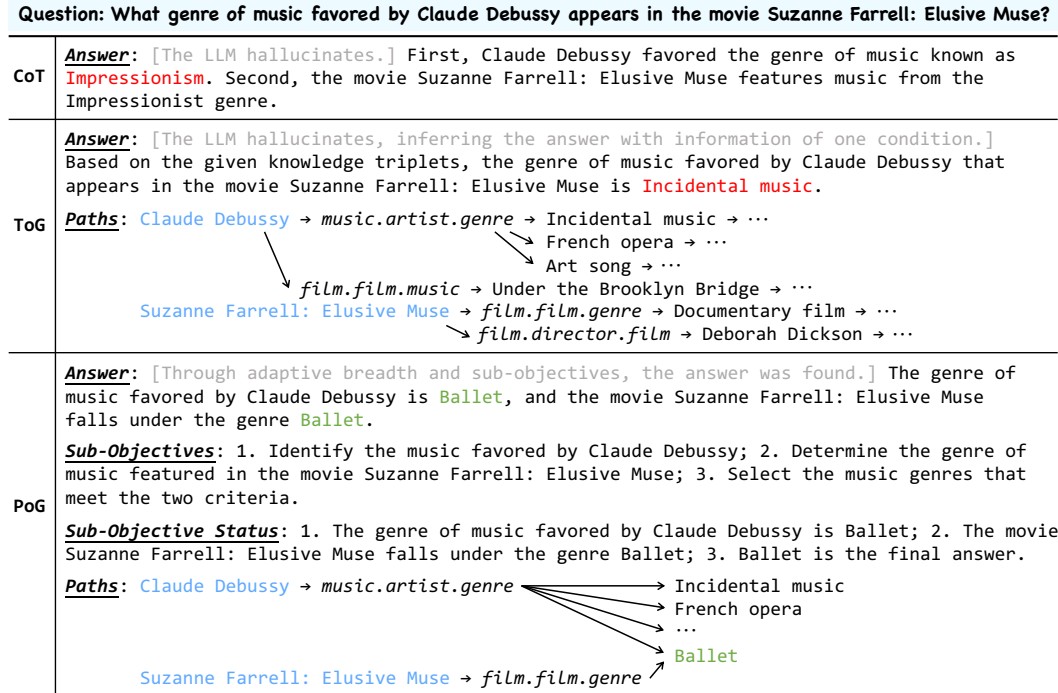

Figure 7: A typical case to compare different methods to answer the complex question. For the convenience of display, we only provide the sub-objective status and partial reasoning paths stored in memory. Topic entities, wrong answers, and correct answers are highlighted in blue, red, and green.

most relevant relation "music.artist.genre" to the topic entity "Claude Debussy". Without constraining the breadth of reasoning paths, PoG considers multiple candidate entities as potentially relevant to the question. In the subsequent exploration of the topic entity "Suzanne Farrell: Elusive Muse", PoG adaptively chooses only "Ballet" as the relevant entity, as it records the known information of sub-objective #1 in memory. Through adaptive breadth and memorization of sub-objective status, PoG successfully and efficiently provides the correct answer. In contrast, ToG randomly explores paths when faced with multiple candidate entities, only finding one condition from sub-objective #1. Finally, ToG only remembers the genre of music favored by "Claude Debussy" but forgets the condition from sub-objective #2, answering "Incidental music". CoT directly hallucinates an irrelevant answer, "Impressionism". This case indicates the effectiveness of adaptive breadth and memorization of sub-objective status.

## H  Broader Impact & Limitation

In the current research landscape, PoG carries a significant broader impact, primarily reflected in its enhancement of complex reasoning capabilities for KG-augmented LLM. By innovatively integrating guidance, memory, and reflection mechanisms, PoG not only strengthens the model's flexibility and accuracy when facing complex queries but also enhances its ability to self-correct erroneous reasoning paths. This self-correcting adaptive planning paradigm enables the model to backtrack and adjust reasoning directions when faced with invalid initial assumptions or impasses, resulting in an optimal solution search. Additionally, the broader impact of PoG is manifested in several other aspects: (1) Improving Efficiency and Effectiveness in Problem-Solving: By dynamically adjusting exploration breadth and employing self-correction mechanisms, PoG can more efficiently handle complex questions and provide more accurate answers, significantly enhancing the overall performance of KGQA systems. (2) Enhancing the Robustness and Adaptability of LLMs: Through its memory mechanism, which records and tracks the completion status and reasoning paths of each sub-objective, PoG enables the LLM to more robustly deal with the uncertainty and complexity of questions, making it more precise and reliable across a wide range of applications. (3) Fostering Innovation in the Field of Artificial Intelligence: PoG's integration of meta-cognitive capabilities into

reasoning and planning processes represents an innovative attempt that could further propel research and innovation in broader AI technologies within the artificial intelligence field. (4) Improving User Experience and Expanding Application Domains: With PoG's reasoning capabilities, user experience is greatly improved due to more accurate and quicker responses. Meanwhile, the domains where it can be applied will also expand, particularly in environments that require handling complex queries and responses involving large volumes of data.

There are still limitations in using PoG for addressing more complex problems. Some of the key limitations include: (1) Low Self-Confidence: LLMs are still not entirely certain about what information is needed, how many steps are required to extract the information, when to perform dynamic updates, or if the current information is sufficient. In future work, we will focus on the evaluation of LLM's self-confidence. For example, this can be alleviated by training a small model specifically for this evaluation task to improve accuracy. (2) Efficiency: Answering complex questions requires multiple steps. In future work, we aim to design strategies to reduce steps and improve task execution efficiency in situations of high self-confidence. (3) Non-Standardized Query: For less standardized queries, semantic understanding might be insufficient due to limitations in the capabilities of the LLM itself, leading to decreased effectiveness. In future work, we will address this issue by employing SOTA query rewriting methods or interacting with the user to refine the query.

