# OpenReview forum: "Plan-on-Graph: Self-Correcting Adaptive Planning of Large Language Model on Knowledge Graphs"
_NeurIPS.cc/2024/Conference — NeurIPS 2024 poster_

### Official Review · Reviewer_Gwnc · 2024-07-04

**Soundness:** 3
**Presentation:** 2
**Contribution:** 2
**Rating:** 6
**Confidence:** 4

**Summary:**

This paper aims to improve KG-augmented LLMs by introducing a self-correcting adaptive planning paradigm. PoG uses three important mechanisms: Guidance, Memory, and Reflection. Guidance leverages LLM to decompose the query into subqueries; Memory stores historical retrieval and reasoning information for reflection; and Reflection uses LLM to decide whether to continue exploration or generate the final output. Experiments show the effectiveness of PoG compared to other SOTA methods and also demonstrate improved efficiency.

**Strengths:**

1. The design of the memory is effective, especially the Sub-Objective Status part. The status of sub-objectives contains the current known information related to the sub-objectives and can effectively remind the LLM of the known knowledge so far and the progress of exploration.
2. The experimental results are also good, showing strong improvement over existing methods.
3. PoG is training-free, which makes it suitable to serve as a plug-and-play for evolving LLMs.

**Weaknesses:**

1. The self-correction is restricted to the retrieval process. However, whether the final answer can be correctedly answer also depend on how the query was decomposed. If the decomposed sub-queries are not suitable for retrieval, the self-correction cannot handle this case.

2. Even though PoG improves efficiency compared with previous methods, it still requires multiple interactions with the LLM (more than 6) and incurs latency, especially for real-time QA (more than 10 seconds on all datasets as reported).

**Questions:**

1. In Table 4, is the number of input/output tokens the average of every single LLM call or for the whole QA?

**Limitations:**

See weakness above

---

> ### Author Rebuttal · Authors · 2024-08-06
>
> Many thanks for your valuable feedback on our paper. We appreciate your recognition of the method design, excellent performance, and generalization. In response to your concerns, we would like to address the following points:
>  - **[W1: Query decomposition]**: Thank you for your constructive suggestion. We fully agree that refining query decomposition is crucial. For more complex queries, we can further improve our model by adapting SOTA query decomposition methods. Still, we would like to clarify that this research direction is orthogonal to our work.
> In this paper, our self-correction focuses on the retrieval process, and our carefully designed decomposition mechanism has shown good results. In future work, we will consider self-correction for query decomposition in KG-augmented LLMs.
>
>  - **[W2: Latency for real-time QA]**: Indeed, there is room for improvement in efficiency based on the results. We would like to clarify that the latency of our method is determined by both the algorithm and the engineering deployment. The current efficiency bottleneck lies in LLM calls and token consumption, which are general limitations in LLM applications. On the algorithmic aspect, where our focus lies, we have significantly reduced LLM calls and token consumption. On the engineering deployment side, there are existing technologies for prompt compression (e.g., LLMLingua) and inference acceleration (e.g., VLLM) that can enhance the system's speed and reduce latency. Recently, there have been increasing explorations of real-world applications based on LLM prompt engineering. We believe that in the future, there will definitely be more universal and efficient methods to improve the efficiency of LLM inference and multi-round interactions.
>  - **[Q1: Token number]**: The number of input/output tokens is the average across all LLM calls for the whole QA. In lines 283-284, we explained that the data statistics pertain to answering a single question.
>
> We hope these responses effectively address your concerns. We will make revisions to further clarify these aspects in our revised paper.

---

> > ### Comment · Reviewer_Gwnc · 2024-08-08
> > **Response**
> >
> > Thank you to the authors for addressing my questions. I will maintain my positive score.

---

> > > ### Author Response · Authors · 2024-08-08
> > >
> > > We greatly appreciate your positive comments and acknowledgment of our paper! In the final version, we will carefully incorporate these clarifications.

---

### Official Review · Reviewer_vd15 · 2024-07-12

**Soundness:** 3
**Presentation:** 4
**Contribution:** 2
**Rating:** 5
**Confidence:** 4

**Summary:**

This paper proposes a new self-correcting adaptive planning paradigm for KG-augmented LLM named Plan-on-Graph (PoG). It has three important mechanism: Guidance, Memory and Reflection. Experiments on three knowledge graph question answering datasets show good results.

**Strengths:**

1, The paper is presented well. The four steps of the algorithm are well explained with Figure 1 and Figure 2, which makes the paper easy to understand. The authors also provide a case study to explain the new method.

2, The experiment results are good compared with other Prompting KG-Augmented LLM methods.

**Weaknesses:**

The weakness of this paper is from the novelty side. The proposed method PoG is like a trivial improvement on the knowledge graph reasoning task based on [Graph of Thoughs (GoT)](https://ojs.aaai.org/index.php/AAAI/article/view/29720). The path explored in this paper also forms a graph like the graph in GoT paper. And the self-reflection and evaluation are like the backtracing and refining steps in GoT paper. The proposed method just adapts these GoT steps in knowledge graph reasoning task.

**Questions:**

Can the authors explain more about the difference of the proposed PoG method and GoT method? My suggestion is to clarify more on why PoG is not a trivial adaption of GoT on knowledge graph tasks.

**Limitations:**

Yes  the authors adequately addressed the limitations.

---

> ### Author Rebuttal · Authors · 2024-08-06
>
> Thank you very much for your valuable feedback on our paper. We appreciate your recognition of the good presentation, excellent performance, and method design. In response to your concerns, we would like to address the following points:
>
>  - **[W1 & Q1: Difference with GoT]**: We would like to emphasize that our method is very different from GoT. Although they all have the structure of a graph, the main difference between GoT and our proposed PoG lies in the ***source of information***: GoT primarily relies on the internal knowledge of the LLM, i.e., the information learned during the pre-training process, while our approach integrates KGs and can utilize additional domain knowledge.
> This core difference results in GoT and other XoT techniques like CoT and ToT being suitable only for enhancing the problem-solving abilities of LLMs that have been covered in pre-training, such as mathematical and logical reasoning problems, but unable to address questions requiring new domain knowledge. To tackle this issue, we propose a novel PoG for KG-augmented LLMs, which has clear distinctions from GoT in the following specific aspects:
>
>    - The roles of the graphs. In our work, the KG is a pre-built external knowledge base, where each node represents an entity and the edges represent relationships between those entities, with the KG serving as ***input*** for the task. In GoT, the reasoning process of an LLM is modeled in graph form, where each node represents a thought and the edges depict relationships between thoughts, with the graph being the ***output***.
>
>    - The methods and objects for evaluation. Our evaluation involves using the LLM to determine whether the information in the current memory can answer the question, which is a ***global*** evaluation. In contrast, GoT scores specific thought nodes on the graph or ranks the nodes to assess whether the LLM thoughts represented by the nodes meet potential correctness criteria, which is a ***local*** evaluation. Besides, the evaluation in GoT is highly sensitive to prompts, requiring different constructions depending on the specific application.
>
>    - The contents of self-correction. We correct the agent's ***navigational process on the KG*** based on the information stored in memory and its own knowledge. The refine operation in GoT refers to looping over a specific thought node on the thought graph, i.e., correcting the ***content of specific thought***, not the navigation on the graph.
>
>    - The destinations of backtracking. We backtrack to ***entities*** in the KG that are related to the query. GoT backtracks to a ***specific thought node*** on the thought graph.
>
>
> We hope these responses effectively address your concerns. We will make revisions to further clarify these aspects in our revised paper.

---

> > ### Comment · Reviewer_vd15 · 2024-08-13
> > **Official Comment by Reviewer vd15**
> >
> > Dear authors,
> >
> > Thanks for your answer. Now I understand more about the innovation part of this paper. I will keep the current score.
> >
> > Thanks,
> >
> > Reviewer vd15

---

> > > ### Author Response · Authors · 2024-08-13
> > >
> > > Thank you very much for acknowledging our paper! We will carefully incorporate these clarifications and further improve the quality of our paper.

---

### Official Review · Reviewer_QT9p · 2024-07-12

**Soundness:** 3
**Presentation:** 3
**Contribution:** 2
**Rating:** 5
**Confidence:** 4

**Summary:**

This paper proposes a self-correcting adaptive planning paradigm for KG-augmented LLM called PoG. It consists of four components: task decomposition, path exploration, memory updating, and evaluation. Experiments on three datasets demonstrate the effectiveness of PoG, outperforming previous methods.

**Strengths:**

1.	The introduction of the self-correcting and adaptive planning mechanism enhances the reasoning capabilities of KG-augmented LLMs.
2.	The experimental results support the effectiveness and efficiency of PoG. The comparison with state-of-the-art baselines demonstrates its superior performance.

**Weaknesses:**

1.	Although the paper has achieved certain results in the experiments, it lacks sufficient analysis on why PoG can achieve better results compared to the baseline models. Additionally, for other KG-augmented LLMs such as ToG[1], ARI[2], and RoG[3], the paper does not sufficiently highlight the advantages and differences of the proposed method.
2.	There is a concern about the efficiency of the proposed method. In a KG, there may be thousands or even millions of entities. Taking Figure 2 as an example, a singer may have hundreds of songs. When the LLM completes the first step of reasoning (finding the singer's songs) and proceeds to the second step of finding the songs that have won awards, the LLM may need to be called hundreds of times? This could potentially lead to inefficiency and increased computational costs.
3.	The paper could provide more details on the potential limitations and challenges of PoG, as well as possible strategies to overcome them.

[1] Think-on-graph: Deep and responsible reasoning of large language model with knowledge graph
[2] Temporal knowledge question answering via abstract reasoning induction
[3] Reasoning on graphs: Faithful and interpretable large language model reasoning

**Questions:**

1.How are the topic entities in the question obtained?
2. When there are too many candidate entities, how does the model ensure the efficiency of reasoning?
3. After selecting the wrong relation or entity, how does the model perform backtracking and reflection?

**Limitations:**

The paper could provide more details on the potential limitations and challenges of PoG.

---

> ### Author Rebuttal · Authors · 2024-08-06
>
> Thank you very much for your valuable feedback on our paper. We appreciate your recognition of the method design, well presentation, and superior performance. In response to your concerns, we would like to address the following points:
>
>  - **[W1: Result analysis]**: Thank you for your constructive suggestion.
> Here, we provide a more detailed analysis of the results as follows:
>
>    - ToG explores reasoning paths with a fixed exploration breadth and cannot detect or correct the errors, showing the limitations in effect and efficiency. We specially design self-correction and adaptive planning mechanisms, which can effectively improve both performance and efficiency.
>    - ARI focuses on QA task for temporal KGs, whereas our work targets KG-augmented LLM task. ARI requires providing past answers and performing historical instance classification on historical information. In contrast, our method does not require historical answers or classification. By utilizing self-correction and adaptive breadth mechanisms to navigate the KG directly, we are able to achieve good performance and efficiency.
>    - RoG requires training and utilizes considerable computational resources and time for planning and reasoning optimization, while our method is training-free. Moreover, the three mechanisms we design (guidance, memory, and reflection) allow PoG's effect to surpass RoG.
>
>     We will add this analysis in the revised paper.
>  - **[W2 & Q2: Excessive LLM calls]**: We would like to clarify that there is no need for hundreds of calls. For numerous candidate entities, we initially use a small model, Sentence Transformer, to calculate the similarity between entities and the question for recall. In contrast, the baseline ToG randomly selects candidates, which ignores semantic information and can easily result in incorrect recall.  It is noted that our method reduces LLM calls by at least 40.8\%.
>
>  - **[W3: More limitations and challenges]**: Many thanks for your valuable advice. Here, we provide more limitations/challenges and the possible solutions as follows:
>    - For less standardized queries, semantic understanding might be insufficient due to limitations in the capabilities of the LLM itself, leading to decreased effectiveness. We can address this issue by employing SOTA query rewriting methods or interacting with the user to refine the query.
>    - LLMs have limited capability in judging whether the available information is sufficient, potentially leading to incorrect answers when information is inadequate. This can be alleviated by training a small model specifically for this evaluation task to improve the accuracy of assessments.
>
>     We will add these contents in the revised paper.
>
>  - **[Q1: Topic entity]**: In our datasets, topic entities are pre-annotated. Please refer to lines 105-107.
>   When such annotations are not provided, the typical approach in the KG domain is to first use named entity recognition and then entity linking methods to identify the topic entities.
>  - **[Q3: Reflection & Backtracking]**:
>   For the details about reflection and backtracking, we introduce them as follows:
>       - Reflection: PoG utilizes the LLM to reflect on whether to correct the current exploration direction based on sub-objective states in $S$ and entities planned for the next iteration of retrieval in $E^D$ (defined in line 140) from memory. Specifically, the LLM assesses whether it is necessary to add entities beyond entities in $E^D$ on current reasoning paths. Please refer to the descriptions of the steps for the reflection in lines 196-200.
>       - Backtracking: PoG leverages the LLM to decide which entities in $E_{cand}$ (defined in lines 159-161 and 201) to backtrack to based on sub-objective states in $S$ and the reason for additional retrieval obtained from the reflection. Specifically, all candidate entities are provided to the LLM, and the LLM selects the specific entities to backtrack to. Please refer to the descriptions of the steps for the backtracking in lines 200-203.
>
>    The specific prompts for these two processes can be found in Appendix A.4.2. Additionally, in lines 308-313, we explained how backtracking and reflection are applied in the specific case shown in Figure 3.
>
>
> We hope these responses effectively address your concerns. We will make revisions to further clarify these aspects in our revised paper.

---

> > ### Comment · Reviewer_QT9p · 2024-08-08
> > **Response**
> >
> > The author's response partially addressed my concerns. I have raised my score. I hope the author further revises the manuscript to make it more complete.

---

> > > ### Author Response · Authors · 2024-08-08
> > >
> > > Thank you very much for acknowledging our paper! We will carefully incorporate these clarifications and further improve the quality of our paper.

---

### Official Review · Reviewer_uH9F · 2024-07-14

**Soundness:** 3
**Presentation:** 3
**Contribution:** 3
**Rating:** 6
**Confidence:** 4

**Summary:**

The paper introduces Plan-on-Graph (PoG), a new paradigm for integrating LLMs with KGs to enhance their reasoning capabilities. The main innovation lies in PoG’s self-correcting adaptive planning mechanism, which addresses the limitations of existing KG-augmented LLMs that rely on predefined exploration spaces and unidirectional navigation. PoG breaks down complex questions into sub-objectives, then iteratively explores and corrects reasoning paths using a combination of Guidance, Memory, and Reflection mechanisms. The Guidance mechanism helps in decomposing the question, Memory stores historical data to support reasoning, and Reflection allows the model to self-correct erroneous paths. The authors validate PoG’s effectiveness through extensive experiments on three KGQA datasets, showing that it outperforms state-of-the-art methods in terms of both accuracy and efficiency.

**Strengths:**

- novel reasoning framework with adaptive searching and reflective thinking.
- great performance on 3 real-world KGQA datasets validate the effectiveness and efficiency of PoG. PoG achieves the best performance across all three datasets, outperforming both LLM-only and KG-augmented LLM baselines.

**Weaknesses:**

- The PoG framework’s complexity, with its multiple components and iterative processes, might make it challenging to implement and optimize.

- The necessity for extensive prompt engineering and management of memory and reflection mechanisms could be resource-intensive.

- While PoG performs well on the evaluated datasets, its generalization to other domains or types of KGs is not thoroughly explored.

- Not compared with other RAG methods such as REPLUG[1].

- The efficiency and cost can be an important limitation. Efficiency in terms of computational resources and time is a concern, as PoG, while efficient, is still resource-intensive. The reliance on KGs means that the model’s knowledge can become outdated if the KG is not regularly updated.




[1] REPLUG: Retrieval-Augmented Black-Box Language Models

**Questions:**

- How does the choice of the exploration depth (set to 4) impact the performance across different datasets? Have you experimented with varying this parameter?

- How does PoG perform with larger and more complex KGs that were not part of the evaluation datasets?

- How does PoG compare with other recent advancements in RAG methods?

- How do you decide when to use the PoG and when to use regular LLM call? Given the expensive cost of PoG, I think it is very unrealistic to run all user queries with PoG. I'd suggest the authors discuss how PoG can be applied in real-world application scenarios.

**Limitations:**

discussed

---

> ### Author Rebuttal · Authors · 2024-08-06
>
> Many thanks for your valuable feedback on our paper. We appreciate your recognition of the novel framework, great performance, and extensive experiments. In response to your concerns, we would like to address the following points:
>
>  - **[W1: Complexity of implementation and optimization]**: Regarding your concern about complexity, we would like to clarify that we have implemented optimizations and improvements regarding efficiency such as adaptive exploration breadth and self-correction. As shown in Table 4, PoG presents ***a speedup of over 4 times*** on the CWQ dataset. We will continue to explore efficiency optimizations in the future. To ensure reproducibility, we will make the implementation of our method open-source.
>
>  - **[W2: Resource consumption]**: We would like to clarify that our method is actually resource-efficient. Firstly, our approach is training-free, which inherently conserves resources. Typically, methods that incorporate KG's knowledge include training and training-free methods. Training methods need to fine-tune LLMs on nearly 400GB of Freebase data, which demands a substantial amount of resources. In contrast, in our method, the resource consumption for prompt engineering, and managing memory and reflection mechanisms, can be gauged from the input token counts listed in Table 4. Compared to the prompting baseline, even though PoG includes several components, PoG shows a reduction of approximately 4.6% in input token consumption.
>
>  - **[W3 & Q2: Generalization]**: Regarding generalization, we would like to explain that Freebase is a suitable dataset for evaluating the generalization of the proposed method for the following reasons:
>    -  Among the most authoritative KGs and test sets used for this task, Freebase has been recognized as complex and large by previous works and includes 1.9 billion triples.
>    -  It covers a wide range of diverse domains such as movies, geography, business and sports. Correspondingly, datasets contain questions about different domains of knowledge, and we can retrieve different subgraphs from Freebase as smaller domain KGs.
>
>    To our knowledge, there currently exists no larger-scale, more complex KG dataset with high-quality question-answer pairs. However, we fully agree that the larger and more complex the KGs we use for evaluation, the more convincing our demonstration of the method's effectiveness will be. In the future, we will explore constructing a high-quality dataset that includes both KGs and corresponding QA pairs spanning diverse domains or types.
>
>  - **[W4 & Q3: Comparison with RAG methods]**: Thank you for your valuable suggestion. We would like to explain that the core idea of RAG involves constructing an index of chunks from documents, and then recalling content based on the similarity between the query and the index. This recalled content provides additional knowledge to the LLM. However, such RAG methods are not designed for understanding graph-structured information and cannot dynamically navigate the KG for KG-augmented LLMs.
> We followed previous works like ToG and RoG, and did not compare with RAG methods.
> For a better explanation, we will include a discussion in the revised paper addressing the distinctions between our method and RAG methods. Additionally, we will cite the REPLUG paper to provide further context.
>
>  - **[W5: Outdated knowledge in KGs]**:
>   We fully agree that the update of KGs is an important topic, and there is a specialized research field dedicated to solving this problem [1].
>   In this paper, our focus is on leveraging KGs to enhance LLM's reasoning, rather than on updating the KGs.
>
>    [1] KartGPS: Knowledge Base Update with Temporal Graph Pattern-based Semantic Rules. ICDE 2024.
>
>  - **[Q1: Depth sensitivity]**:
>  We have chosen the proper setting based on our preliminary experiments. Specifically, the results on CWQ dataset are presented as follows:
>     | Depth | 1    | 2    | 3    | 4    | 5    |
>     |-------|------|------|------|------|------|
>     | PoG   | 53.9 | 56.6 | 61.7 | 63.2 | 64.4 |
>
>     This indicates that increasing the depth improves performance. Beyond a depth of 4, the improvement becomes less noticeable. Therefore, we set the depth to 4.
>  - **[Q4: Selection of usage]**:
>   Thank you for your valuable suggestion. The current best practice in real-world applications is query routing, which involves designing a router to determine whether to call an agent or a regular LLM. This practice has been successfully implemented in many popular projects like LlamaIndex and LangChain. These routers can also be applied to our PoG in case of real-world deployment. We will add this discussion in the revised paper.
>
> We hope these responses effectively address your concerns. We will make revisions to further clarify these aspects in our revised paper.

---

> > ### Comment · Reviewer_uH9F · 2024-08-13
> >
> > Thank you for your response and clarification. I have also read the comments from other reviewers and authors' response. I'd like to keep my original rating.

---

> > > ### Author Response · Authors · 2024-08-13
> > >
> > > Thank you very much for acknowledging our paper! We will carefully incorporate these clarifications and further improve the quality of our paper.

---

### Official Review · Reviewer_kA7J · 2024-07-17

**Soundness:** 2
**Presentation:** 2
**Contribution:** 3
**Rating:** 5
**Confidence:** 4

**Summary:**

This paper introduces Plan-on-Graph (PoG), a novel self-correcting adaptive planning paradigm for Knowledge Graph-augmented Large Language Models (KG-LLM). PoG aims to address limitations in existing KG-augmented LLM approaches by decomposing questions into sub-objectives and iteratively exploring reasoning paths, updating memory, and reflecting on the need for self-correction. The method incorporates three key mechanisms: Guidance, Memory, and Reflection, designed to ensure adaptive breadth in self-correcting planning for graph reasoning. Experiments on three real-world datasets demonstrate the effectiveness and efficiency of the proposed approach.

**Strengths:**

- Novel approach: The paper introduces reflection and self-correction in KG path exploration for KGQA, addressing limitations of fixed path breadth and irreversible exploration in previous methods.
- Comprehensive context: The paper provides a clear and organized discussion of related work in KG-augmented LLMs, offering a good overview of the research area.
- Experimental comparison: The study includes different types of baselines in the experiments, demonstrating the improvements achieved by the proposed model in a broader context.

**Weaknesses:**

- Missing important implementation details: While the paper provides visualized knowledge subgraphs to represent path exploration and memory updates, there is no explicit template or example showing how this graph or tree structure is exactly flattened in the prompts of the LLM agent. The use of arrows for paths in both Figure 3 and 5 creates ambiguity about the real implementation. This makes reader unclear and concerns about the real implementation.

- Concerns about model dependency: The proposed PoG method is a pure prompting approach, which raises questions about its applicability and performance on smaller open-source models. There are concerns about whether the performance improvements achieved by PoG are largely due to the capabilities of the advanced base models used, rather than the method itself.


- Lack of critical evaluations: The paper is missing several important analyses, such as:
  - The proportion of cases where the LLM needs to trigger reverse direction path visiting
  - The frequency with which this revisiting leads to correct or incorrect answers
  - How the number of breadth in each step affects the effectiveness of the proposed method
  - Whether the length of hops from the query entity to the answer entity affects the number of revisiting and the method's effectiveness
  - How consistent or diverse is the exploration path under the proposed method if experiment with multiple trails?

- Minor thoughts: Additionally, it would be interesting if the authors could discuss or explore whether this method of constructing paths and memories could support effective post-training of open-source models, though this is not a critical weakness.

**Questions:**

Please refer to the questions raised in Weakness section.

**Limitations:**

Yes, the author has discussed the limitations in Appendix G, and leave them as future work.

---

> ### Author Rebuttal · Authors · 2024-08-06
>
> Thank you very much for your valuable feedback on our paper. We appreciate your recognition of the novel approach, comprehensive context, and extensive experiments. In response to your concerns, we would like to address the following points:
>
>  - **[W1: Missing prompt details & Implementation ambiguity]**:
>     Thank you for your valuable suggestions. Here, we provide two explicit examples to show how the KG is flattened in the prompts.
>
>     - Relation Exploration:
>         ```
>       Q: Name the president of the country whose main spoken language was Brahui in 1980?
>       Sub-objectives: ['Identify the countries where the main spoken language is Brahui', 'Find the president of each country', 'Determine the president from 1980']
>       Topic Entity: Brahui Language
>       Relations: language.human_language.main_country; language.human_language.language_family; language.human_language.iso_639_3_code; base.rosetta.languoid.parent; language.human_language.writing_system; base.rosetta.languoid.languoid_class; language.human_language.countries_spoken_in; kg.object_profile.prominent_type; base.rosetta.languoid.document; base.ontologies.ontology_instance.equivalent_instances; base.rosetta.languoid.local_name; language.human_language.region
>       Output: ['language.human_language.main_country', 'language.human_language.countries_spoken_in', 'base.rosetta.languoid.parent']
>         ```
>
>     - Memory Update:
>         ```
>         Q: Find the person who said "Taste cannot be controlled by law", what did this person die from?
>         Sub-objectives: ['Search the person who said "Taste cannot be controlled by law"', 'Search the cause of death for that person']
>         Memory: {
>             "1": "No information.",
>             "2": "No information."
>         }
>         Knowledge Triplets: Taste cannot be controlled by law. media_common.quotation.author [Thomas Jefferson]
>         Output: {
>             "1": "Thomas Jefferson said 'Taste cannot be controlled by law'.",
>             "2": "It is not mentioned, and I also don't know."
>         }
>
>         ```
>
>     We will add these details about prompts in our revised paper. Regarding the ambiguity in Figure 3 and 5, we will improve the description and presentation of figures. Additionally, to ensure reproducibility, we will release our full project including the code and prompts after this paper is published.
>
>
>  - **[W2: LLM dependency]**: For your concern about the model dependency, we would like to emphasize that our method can be applicable to any LLM and we focus on how to utilize KGs to enhance LLMs by proposing a novel KG-augmented LLM paradigm which is model-agnostic.
>   Our goal is to enhance the reasoning capabilities of LLMs, rather than simply improving the performance of KGQA. In our experiments, we compared the KG-augmented LLM approaches with the LLM-only approaches using the same LLMs, validating the effectiveness of our method in enhancing the reasoning capabilities of LLMs.
>
>  - **[W3: Lack of evaluations]**: Thank you very much for your valuable and constructive suggestions. Regarding your mentioned evaluations, we respond to each as follows:
>    - *The proportion of cases where the LLM needs to trigger reverse direction path visiting*: 24% on CWQ dataset.
>    - *The frequency with which this revisiting leads to correct or incorrect answers*: The results are presented in Appendix F (Figure 4), i.e., 48%, 64%, and 36% correct answers on three datasets.
>    - *How the number of breadth in each step affects the effectiveness of the proposed method*: We would like to clarify that the breadth in each step is adaptively chosen by the LLM, without any fixed setting. The detailed descriptions can be found in lines 144-146, 152-154, 162-164, and Appendix A.2.
>    - *Whether the length of hops from the query entity to the answer entity affects the number of revisiting and the method's effectiveness*: The results on CWQ dataset are presented as follows:
>
>         | Depth | 1    | 2    | 3    | 4    | 5    |
>         |-------|------|------|------|------|------|
>         | PoG   | 53.9 | 56.6 | 61.7 | 63.2 | 64.4 |
>
>       This indicates that increasing the depth improves performance. Beyond a depth of 4, the improvement becomes less noticeable. Therefore, we set the depth to 4.
>
>    - *How consistent or diverse is the exploration path under the proposed method if experiment with multiple trails*: In applications of LLMs, the consistency of LLM's output from multiple runs depends on the temperature setting, which can be adjusted according to the requirements. If complete consistency is required, the temperature can be set to 0. In other cases, if diversity is needed, a higher temperature setting can be used to obtain more diverse results. As described in Appendix E, we set the temperature to 0.3 in our experiments. The results show that the changes in exploration paths are very small.
>
>     In the revised paper, we will add supplemented evaluations on all datasets.
>
>  - **[W4: Discussion on fine-tuned LLMs]**: Many thanks for your valuable advice. We would like to clarify that our proposed method is training-free, which is one of our advantages. From existing works, it is evident that training incurs high costs, and a training-free method offers a more cost-effective solution. Due to the short rebuttal period, we are currently unable to provide training results for the open-source LLM. In future work, we will explore whether constructing paths and memory in a post-training condition is more effective, and whether the current issue of LLMs' insufficient understanding of KG structures can be addressed through training.
>
>
> We hope these responses effectively address your concerns. We will make revisions to further clarify these aspects in our revised paper.

---

> ### Author Response · Authors · 2024-08-11
>
> Dear Reviewer kA7J,
>
> We would like to express our sincere gratitude for the time and effort you spend reviewing our paper. As **the author/reviewer discussion stage draws to a close**, we are eager for your response to ascertain if our detailed response has sufficiently addressed your concerns. We would be honored to address any further questions you may have. *We eagerly anticipate and highly value your re-evaluation of our paper.*
>
> Thank you once again for your thorough review of our paper.
>
> Best regards,
>
> Authors of Submission 4240

---

> ### Comment · Reviewer_kA7J · 2024-08-11
>
> Thanks the authors for the rebuttal, especially the additional prompting details and analytical evaluations well address my original concern, please consider adding these important details into future editions of the paper. Regarding the LLM dependency and potential discussion of post-training, I acknowledge and understand the proposed method is training-free as this is a pure prompting approach. My original concern is raised based on the relatively lower reasoning and instruction-following ability provided by most open-sourced smaller LLMs, it would be definitely beneficial if the proposed method could exhibit enhancement on these models. Overall, I appreciate the work and the responses, I have raised my score. Thanks.

---

> > ### Author Response · Authors · 2024-08-12
> >
> > Thank you very much for your feedback and for acknowledging our work. We appreciate your suggestions and will definitely consider adding more details to further enhance our paper in future editions. We are grateful for the score increase!

---

### Decision · Program_Chairs · 2024-09-25

**Decision:**

Accept (poster)

**Comment:**

In this work, the authors propose Plan-on-Graph (PoG), a self-correcting adaptive planning paradigm, which aim to help KG-augmented LLM systems to solve question answering task. Given a question, the pipeline first decomposes the question into a sequence of sub-objectives; then it repeats the loop of 1) reasoning path exploration in the KG; 2) updating useful information into a memory; and 3) reasoning to find an answer if with sufficient inforamtion or reflect for other reasoning paths otherwise. The authors tested their method on three KGQA tasks, they show PoG outperform a set of baseline systems on all tasks.

This work has many merits, the all-positive scores from the five reviewers reflect this. All reviewers agree that the work is well motivated as self-correction and adaptive planning seem to enhance LLM system's reasoning capabilities; applying such idea and technique in KG path exploration for KGQA is novel; the presentation is clear, the paper is well structured, especially the related work section; the experiment design is good, experimental results support the claim well; the fact that the method is training-free makes it easy to plug-and-play.

On the negative side, reviewers' main concerns were around the following aspects:
- Unclear how PoG connects/compares with prior works that also utilize graph-structured memory, such as Graph-of-Thought.
- Unclear about efficiency, latency and cost limitation because PoG (as a inference-only method) may require huge amount of API calls when the question is compositional complex or the KG is huge.
- Unclear about how PoG could be applied using more accessible models as backbone.
- Missing implementation details.

Reviewers agree that in general the authors did a good job addressing and clarifying the concerns, two reviewers have increased their score from 4 to 5. However, there are a few points remain less satisfied:
- The authors suggest that their proposed paradigm is model-agnostic, their goal is to enhance the reasoning capabilities of LLMs, rather than simply improving the performance of KGQA. This is only partially true, because there lacks of evidence the PoG could be directly applied to weaker/open-sourced LLM systems. What is the minimal requirements on the backbone end to make this working?
- In multiple steps of the pipeline, the LLM is asked to generate an output based on the provided information (e.g., retrieved from the KG) and `its own knowledge`. The fail of disentangling external knowledge and the LLM's embedded knowledge makes it less straightforward to understand the proposed method.
- It is only in the appendix that the authors say they are using in-context learning in all the prompts (Appendix A). Specifically, they use 5-shots learning on all datasets (Appendix E). They fail to provide any detail on how the five examples are being selected/retrieved, are they using the same in-context examples across datasets, across data points, or per data point? This makes it even more difficult to fully understand how the proposed method works.
- Related to the above points, although the authors provide a good ablation study showing how each components could affect the overall system performance, they fail to discuss deeper how different design of each component could make a difference. Because PoG is a training-free method, specific prompt design, including the way to obtain the in-context examples could be rather important. This once again is related to how accessible the method is, because different backbone model may need very different prompt designing/engineering practice.

Given the all-positive scores, I tentatively recommend accept, but because:
- there are still some critical points unclear (see my review above); and
- all reviewers end up giving lukewarm scores,

I will not fight for acceptance if SAC suggests otherwise.